# An AlScN Piezoelectric Micromechanical Ultrasonic Transducer-Based Power-Harvesting Device for Wireless Power Transmission

**DOI:** 10.3390/mi15050624

**Published:** 2024-05-06

**Authors:** Junxiang Li, Yunfei Gao, Zhixin Zhou, Qiang Ping, Lei Qiu, Liang Lou

**Affiliations:** 1School of Microelectronics, Shanghai University, Shanghai 201800, China; ljx62458@shu.edu.cn (J.L.); gaoyunfei@shu.edu.cn (Y.G.); zzx1999@shu.edu.cn (Z.Z.); 2The Shanghai Industrial μTechnology Research Institute, Shanghai 201899, China; 3School of Electronic and Information Engineering, Tongji University, Shanghai 201804, China; pingq86@tongji.edu.cn (Q.P.); qiulei@tongji.edu.cn (L.Q.)

**Keywords:** PMUT, wireless energy transfer, implantable medical devices, rectifier circuits

## Abstract

Ultrasonic wireless power transfer technology (UWPT) represents a key technology employed for energizing implantable medical devices (IMDs). In recent years, aluminum nitride (AlN) has gained significant attention due to its biocompatibility and compatibility with complementary metal-oxide-semiconductor (CMOS) technology. In the meantime, the integration of scandium-doped aluminum nitride (Al_90.4%_Sc_9.6%_N) is an effective solution to address the sensitivity limitations of AlN material for both receiving and transmission capabilities. This study focuses on developing a miniaturized UWPT receiver device based on AlScN piezoelectric micro-electromechanical transducers (PMUTs). The proposed receiver features a PMUT array of 2.8 × 2.8 mm^2^ comprising 13 × 13 square elements. An acoustic matching gel is applied to address acoustic impedance mismatch when operating in liquid environments. Experimental evaluations in deionized water demonstrated that the power transfer efficiency (PTE) is up to 2.33%. The back-end signal processing circuitry includes voltage-doubling rectification, energy storage, and voltage regulation conversion sections, which effectively transform the generated AC signal into a stable 3.3 V DC voltage output and successfully light a commercial LED. This research extends the scope of wireless charging applications and paves the way for further device miniaturization by integrating all system components into a single chip in future implementations.

## 1. Introduction

Implantable medical devices (IMDs) have become increasingly important in monitoring and treating various diseases due to the rapid advancement of modern medical technology. IMDs are designed to monitor biosignature parameters, facilitate drug delivery, or enhance the functionality of human organs. IMDs are expected to become essential medical tools for monitoring patients’ health, particularly benefiting individuals with severe conditions such as cardiovascular diseases, neurological disorders, and diabetes mellitus [1,2,3,4,5,6]. Since the 1950s, there have been many technological advancements, including low-power electronics, interfaces, RF electronics, micromechanical sensors, and miniature batteries. Technological advancements offer opportunities for designing and realizing sophisticated implantable devices [7].

Historically, conventional batteries have served as the primary energy source for most implantable medical devices, especially those necessitating high energy density and prolonged implantation durations. Nonetheless, the widespread adoption of conventional batteries has been impeded by their bulky dimensions and limited lifespan. Consequently, batteries integrated into implantable devices often necessitate periodic replacement through surgical procedures, posing economic and physiological burdens on patients [8,9]. Hence, the quest for long-term, stable, and reliable energy sources has become a pressing concern for implantable medical device applications. Various alternative energy sources have been developed in response to the challenges posed by traditional batteries. One of these is wireless energy transmission, which has emerged as a technology of significant importance. It is particularly well-suited for powering low-power devices implanted within the human body, such as pacemakers, artificial retinas, and cochlear implants. Wireless power transmission methods have the potential to downsize existing medical devices, facilitating real-time health monitoring and fostering the advancement of personalized and preventive IMDs. Various techniques of wireless energy transmission are documented in the literature, including mid-field and far-field electromagnetic radiation methods, near-field inductive coupling, and ultrasound-induced wireless energy transmission [10,11]. Ultrasonic wireless power transfer (UWPT) stands out for its ability to deliver more power from a compact device compared to other approaches, which is attributed to its shorter wavelength, which results in a diminutive receiver size. Furthermore, UWPT reduces attenuation in human tissues, enabling deeper penetration [12]. Moreover, ultrasound is widely regarded as safer in diverse medical applications. With a longstanding history in medical diagnosis and treatment, ultrasound technology adheres to safety standards; for instance, the U.S. Food and Drug Administration (FDA) specifies that the permissible ultrasound intensity for medical diagnostic purposes must not exceed 720 mW/cm^2^, significantly higher than that of radio waves (1–10 mW/cm^2^) [8]. Hence, ultrasound-induced radio energy transmission methods have been proven to be well-suited for energizing implantable medical devices. Lead zirconate titanate (PZT) has been widely used and studied in ultrasonic transducers due to its cost-effectiveness and favorable piezoelectric properties [13,14,15,16,17]. However, PZT materials present a potential safety hazard to the human body and are incompatible with complementary metal-oxide-semiconductor (CMOS) processes. Compared to PZT, lead-free aluminum nitride (AlN) is a more biocompatible material [18]. AlN is a promising candidate for low-cost, highly integrated piezoelectric micromachined ultrasonic transducer (PMUT) devices due to its biocompatibility, CMOS compatibility, and decent sensitivity. Consequently, AlN-based PMUTs have recently gained popularity as acoustic transmitters and receivers in various applications. Furthermore, AlScN films demonstrate a more excellent piezoelectric response than that of pure AlN films [19,20,21].

In this study, a wireless power harvesting system using AlScN-based PMUTs is reported, and significant improvement of the power transfer efficiency (PTE) by up to 2.33% is demonstrated. The system’s output produces a stable 3.3 V DC voltage, which can light a commercial LED stably. An acoustic matching gel with an acoustic impedance coefficient similar to that of deionized water was utilized for encapsulation treatment, aiming to reduce acoustic wave reflection and enhance energy utilization efficiency. Additionally, we implemented a Dickson rectifier circuit with controllable output voltage and low output ripple to convert AC voltage into DC voltage. This study proves the feasibility of AlScN PMUTs for power harvesting. In the future, the device can be integrated into a single chip, greatly reducing its size and making it more suitable for implantable applications.

## 2. Ultrasonic Wireless Power Transfer System

Figure 1 shows a block diagram of the UWPT system, which consists of three main components: energy transmission, transmission medium, and energy reception. The energy transmission segment includes a signal generator and transmitting transducer. The transmission medium uses deionized water with an acoustic impedance coefficient similar to that of human tissues. Finally, the energy reception element comprises the receiving transducer and back-end circuit.

The signal generator plays a crucial role in the ultrasonic wireless energy transmission system by providing a frequency-adjustable AC signal to drive the emission transducer. When the transmitting transducer operates in its mechanical resonance state, the piezoelectric material exhibits its maximum automated amplitude through the inverse piezoelectric effect, resulting in the most robust generation of ultrasonic energy. Aligning the AC frequency generated by the signal generator with the resonant frequency of the transmitter transducer enables the optimal conversion efficiency between electrical and acoustic power. This synchronization ensures that the system operates at peak performance levels, maximizing the efficacy of energy transfer.

The acoustic signal is transmitted directionally through the deionized water medium in the ultrasonic wireless energy transmission system. The receiving transducer captures the acoustic signal and utilizes the piezoelectric effect of the piezoelectric material to convert it into an AC electrical signal of identical frequency. Subsequently, the back-end circuit system comes into play, carrying out essential functions such as voltage-doubling rectification, energy storage, and voltage regulation. This circuitry facilitates the conversion of the AC power into a stable DC output voltage, which is then used to power the designated load application effectively.

## 3. Structure, Fabrication, and Characterization of PMUTs

### 3.1. Structure and Simulation of PMUTs

This study uses an ultrasonic transducer (Audiowell, US0072, Guangzhou, China) with PZT as the piezoelectric material for the transmitting transducer (TX). PZT transducers have a high electromechanical coupling coefficient and emission sensitivity, enabling efficient electro-acoustic conversion. These emission transducers are usually placed outside the human body, reducing the risk of lead leakage and ensuring user safety.

The system’s receiver transducer (RX) is a PMUT array that uses AlScN as the piezoelectric material. Figure 2a illustrates the structure of the PMUT, which consists of an upper piezoelectric oscillator and a lower substrate. The upper piezoelectric oscillator is a sandwich structure that includes upper and lower electrodes (Mo), a piezoelectric layer (AlScN), and a vibrating layer (Si). This configuration is crucial for achieving the piezoelectric effect. The lower substrate serves as a support structure with a cavity that aids fixation [22]. Upon receipt of the acoustic signals generated by the TX, the thin film material of the RX’s piezoelectric layer undergoes periodic bending deformation. According to the principle of the piezoelectric effect, the material surface alternately generates positive and negative charges, thereby producing an alternating current signal output.

For the AlScN PMUT at the receiving end, the receiving sensitivity (*G_s_*) of a thin-film piezoelectric MEMS device can be described as [23]
(1)$$G_{s}\propto e_{31,f}/\varepsilon_{33}$$
where *e*_31,*f*_ is the piezoelectric coefficient, and *ε*_33_ is the dielectric constant. Table 1 illustrates the comparison of material parameters between AlScN-doped with 9.6% scandium and pure AlN. The data presented in the table indicate that AlScN exhibits a higher sensitivity than that of the AlN material, implying that AlScN PMUTs can generate a greater charge output when subjected to the same acoustic pressure.

For advanced PMUT design, finite element analysis (FEA) was conducted, and the three-dimensional simulation model is shown in Figure 3a. The first-order vibration mode of a single PMUT is illustrated in Figure 3b. Table 2 presents the geometric parameters of each structural layer of the PMUT.

To further investigate the influence of the piezoelectric layer's thickness on its performance, the simulation results (Figure 3c) indicate that the receiving sensitivity increases with the thickness of the piezoelectric layer up to 2.3 μm. However, as the deposition thickness increases, the difficulty of the etching processes increases. Based on previous processing experience of the Shanghai Industrial μTechnology Research Institute (SITRI), a 1 μm AlScN film has been well verified with good performance. Therefore, a 1 μm thick AlScN film is chosen based on the trade-off between the theoretical optimization and actual processing capability. Subsequently, it was determined that the cavity side length of each PMUT unit should be approximately 180 μm to achieve the designed operating frequency. The optimal performance can be achieved when the electrode radius is approximately 70% of the diaphragm radius, as illustrated in Figure 3d. Based on the processing and testing experience of SITRI, the best performance can be achieved with a side length of approximately 140 μm, 78% of the diaphragm radius, for the top electrode. 

The PMUT array in this paper utilizes a square structure, resulting in a higher fill rate of up to 81%. This allows more PMUTs to be placed in parallel within a limited space, resulting in high output power density and higher receiving sensitivity [25]. The PMUT array has an area of 2.8 mm × 2.8 mm, a cell spacing of approximately 200 μm, and 13 × 13 square elements. 

### 3.2. Fabrication of PMUTs

Figure 4 shows the flow of the PMUT fabrication process, which begins with a customized cavity silicon-on-insulator (CSOI) wafer. (a) The wafer comprises a 4.5 μm silicon device layer with a 1 μm oxide layer buried underneath. (b) An AlScN seeding layer of approximately 100 nm was deposited using atomic layer deposition (ALD) to enhance the c-axis orientation of AlScN and the film quality of the top Mo layer before the deposition of the Mo/AlScN/Mo stack. (c) The AlScN seeding layer was first deposited via magnetron sputtering, followed by a 0.2 μm Mo/1 μm AlScN/0.2 μm Mo layer. (d) Photolithography and dry etching were used to pattern the top Mo layer. (e) Plasma-enhanced chemical vapor deposition (PECVD) was used to deposit the SiO_2_ layer, which was then etched via reactive ion etching (RIE) to open the through-hole to the top electrode. Finally, the bottom-to-top contact hole was opened with AlScN anisotropic dry etching. Finally, the aluminum leads and pads are deposited and patterned. The Sc content of AlScN is 9.6%, which was decided in conjunction with the actual process level. The entire fabrication process of the PMUT was carried out at the Shanghai Industrial μTechnology Research Institute.

The selection of the transducer’s operating frequency requires balancing several interrelated factors. These include energy attenuation in the propagation medium, transducer thickness, natural focal distance, and the size of reactive components in TX and RX, among other losses [26]. The recommended operating frequency range is typically between 200 kHz and 1.2 MHz for transducers used to power implantable biomedical devices. For transmitters, the Rayleigh distance increases with frequency. The Rayleigh distance is the point at which the near-field becomes the far-field. This range is where acoustic energy is most concentrated and is the optimal location for the receiver. Therefore, higher-frequency-transmitting transducers can position the receiving transducer in a deep field, which is necessary for powering devices that need to penetrate deep into the human body (e.g., pacemakers). However, high-frequency ultrasound propagation in the medium also causes energy loss, which is disadvantageous for energy transmission efficiency. We selected a transmitting transducer operating at approximately 1 MHz for this study. This frequency was chosen because the transducer has a longer Rayleigh distance of around 61 mm while ensuring that the sound wave signal is not too severely attenuated in the medium. In a wave-propagating medium, the Rayleigh distance (*L*) depends on the transducer radius (*a*) and the acoustic wavelength (*λ*), as expressed by the following equation [27,28]:(2)$$L=\frac{{\left(2a\right)}^{2}-\lambda^{2}}{4\lambda^{2}}\approx \frac{a^{2}}{\lambda },\;\mathrm{FOR}{\left(2a\right)}^{2}>>\lambda^{2}$$

To ensure a certain level of transmission efficiency, we positioned the receiving transducer in the experiment at a distance of approximately 20 mm, which is shorter than the Rayleigh distance. This reduces acoustic losses through the medium.

### 3.3. Characterization of PMUTs

After processing the PMUT array, we conducted an initial optical surface examination of their morphological features using a 3D measurement laser microscope (OLYMPUS OLS5000, Tokyo, Japan). The findings are depicted in Figure 5. The entire PMUT array measures approximately 2.8 mm × 2.8 mm, comprising a total of 13 × 13 square elements. The cross-sectional scanning electron microscope (SEM) image on the right displays the prepared PMUT. A slight discrepancy between the original design and the fabricated device is visible, attributed to manufacturing errors during fabrication.

This study considers the simulated use scenario of a receiver inside the human body and uses the acoustic impedance coefficient of human soft tissues, which is approximately 1.63 MRayl [29]. To ensure optimal acoustic wave transmission, a non-toxic and non-polluting polyurethane adhesive with an acoustic impedance coefficient of 1.5 MRayl is used as the acoustic matching material for encapsulating the PMUT array. This material choice closely matches human body tissues’ acoustic impedance, improving acoustic wave transmission throughout the system. 

As shown in Figure 6, the resonance frequency (*f_r_*) and anti-resonance frequency (*f_a_*) of the unsealed PMUT array were measured to be 1.997 MHz and 2.01 MHz, respectively, using an impedance analyzer (KEYSIGHT E4799A, Santa Rosa, CA, USA). According to the effective electromechanical coupling coefficient ($K_{eff}^{2}$) equation [30],
(3)$$K_{eff}^{2}=\frac{f_{a}^{2}-f_{r}^{2}}{f_{a}^{2}}$$
the $K_{eff}^{2}$ value of a PMUT in the air is 1.29%. The resonance frequency of the PMUT array sealed with polyurethane adhesive decreases to 1.002 MHz, and the anti-resonance frequency is 1.041 MHz due to the presence of an adhesive that increases the virtual added mass effect [31]. In addition, the PZT transmitter transducer’s resonance frequency was also measured to be 0.922 MHz, the anti-resonance frequency was measured to be 0.993 MHz, and the $K_{eff}^{2}$ value was measured to be 13.79%.

## 4. Experimental Setup and Results

Deionized water has an acoustic impedance coefficient of 1.48 MRayl, similar to that of human soft tissue. However, due to its low attenuation coefficient (0.002 dB/cm MHz), deionized water is limited in accurately simulating soft tissue environments. Additionally, deionized water exhibits a significant temperature dependence, with the speed of sound varying by up to 50 m/s within the temperature range of 20–40 °C [29]. Despite these limitations, deionized water is still commonly used as an alternative to soft tissue for various performance tests on receivers. This choice is primarily driven by its cost-effectiveness, easy availability, and the convenience of freely moving the transducer in a water bath for testing purposes. 

In Figure 7a, the performance of the RX was investigated within a plastic water tank (245 mm × 175 mm × 90 mm) filled with deionized water. The transmitter and receiver transducers were fixed at the bottom of the tank, maintaining a distance of 20 mm between them. An AC signal generator (Keysight 33600A, Sunnyvale, CA, USA) was employed at the transmitter side to produce an AC signal in the frequency range of 0.8 to 1.2 MHz. The signal was amplified by a power amplifier (Falco Systems WMA-300, Katwijk aan Zee, The Netherlands) and transmitted with a peak-to-peak input voltage (Vpp) of 40 Vpp.

An oscilloscope (KEYSIGHT DSOX2014A, Santa Rosa, CA, USA) was used to measure the variation in the Vpp of the PMUT device under different load resistances during an ultrasonic drive across a wide frequency range (0.8–1.2 MHz) (Figure 7b). The results show that the maximum voltage is obtained at 0.98 MHz over the entire test range. Section 3.3 establishes that the TX has the highest electro-acoustic conversion efficiency at 0.922 MHz. In comparison, the RX has the most vital ability to convert acoustic signals into electrical signals at 1.002 MHz (excluding mechanical loss of the transducer). Hence, it is logical that the maximum output voltage frequency falls between the resonant frequencies of the two transducers.

In Figure 7c, the plot illustrates the change in the output voltage signal concerning the variation in the load resistance in series with the RX, ranging from 50 Ω to 1 MΩ. The output voltage amplitude increases as the load resistance increases, eventually reaching a saturation point at higher external loads. Figure 7d shows that the maximum instantaneous power density of the device reaches its peak at 16.44 mW/cm^2^ when the load resistance is approximately 430 Ω. These results suggest that the developed device effectively maximizes electrical power transmission when appropriately matched with the load impedance [32].

This investigation further characterized the relationship among the output power density, transfer efficiency, and input voltage while holding the load resistance constant at 430 Ω. Figure 8a depicts that the output power density increases with the increase in the input voltage when the input voltage is relatively small. However, when the input voltage surpasses 150 Vpp, the output voltage does not exhibit a significant increase. This phenomenon can be attributed to thermal effects and cavitation phenomena induced by energetic ultrasound in liquids, resulting in additional energy loss [33]. Such effects are more pronounced in soft tissues. The transmission efficiency reaches its maximum when the input voltage is 40 Vpp, corresponding to an incident power density of 705 mW/cm^2^. The gray area in Figure 8a indicates compliance with the FDA limit on incident power density.

Additionally, the impact of the rotational angle of the receiver on the output open-circuit voltage was investigated by placing the RX at two near-field distances of 10 mm and 20 mm, as well as at the Rayleigh distance (~61 mm) for testing purposes. As shown in Figure 8b, the rotation angle of the RX exerts a considerable influence on the output voltage, while the distance also affects the output voltage to some extent. When the TX and RX are aligned positively, the sound intensity on the surface of the RX is maximized, resulting in the most significant open-circuit output voltage at that alignment.

## 5. Circuit Simulation and Implementation

### 5.1. Voltage-Doubling Rectifier Circuit

The RX converts acoustic energy into electrical energy through piezoelectric conversion. The resulting output energy is unstable, with relatively weak voltage and current. The alternating frequency matches the input acoustic frequency, making it prone to fluctuations and unsuitable for direct application in microelectronic equipment. To make this energy usable by a load, it must be processed through a rectifier circuit, an energy collection circuit, and a voltage regulator conversion circuit.

While a bridge rectifier structure is commonly used for rectification, its output voltage is typically lower than the input voltage’s amplitude. The resulting DC voltage may not be sufficient to activate the back-end voltage regulator chip. Therefore, this study employs a voltage-doubling rectifier circuit. This circuit, comprising diodes and capacitors, is vital for energy harvesting, as it boosts the captured AC signal and converts it into DC voltage with minimal loss [34,35,36,37,38,39]. Cascading more circuits can increase the voltage across the load but reduce the current flowing through the load. This trade-off can lead to undesirable charging delays in energy storage capacitors. Additionally, the limited PCB board area restricts the number of cascaded circuits that can be implemented.

When dealing with AC signals converted from PMUTs' electromechanical energy, it is advisable to use a diode with the lowest possible turn-on voltage. This is because the peak voltage is typically low and may not be enough to turn on a regular diode. When dealing with AC signals converted from PMUTs' electromechanical energy, it is advisable to use a diode with the lowest possible turn-on voltage. Additionally, diodes with extremely short reverse current recovery times are essential for energy-harvesting circuits that operate at high frequencies [39].

The Schottky diode is distinct from a semiconductor–semiconductor junction in that it is a metal–semiconductor junction. It has an exceptionally low forward turn-on voltage and short reverse current recovery time. This study used an ON Semiconductor Schottky diode, specifically, the 1N5817 model. This diode has a turn-on voltage of approximately 150 mV and a conduction voltage drop of around 0.4 V at a current of 1 A.

In the field of energy harvesting and wireless energy transmission, the most commonly used voltage-doubling rectifier circuits are the Villard voltage doubler (Cockcroft-Walton voltage doubler) and the Dickson voltage doubler. These circuits have structures, as depicted in Figure 9a,b. The output voltage generated by these voltage multipliers is influenced by factors such as the number of diode stages, input voltage, and load resistance. The power transfer efficiency for a given voltage multiplier configuration depends on the input power stage, diode characteristics, number of stages, and load conditions. SPICE software (LT spice 24.0.12) simulations have shown that in the Villard structure, the outputs of the odd nodes differ from those of the even nodes. Precisely, the outputs of the odd nodes are consistently clamped upwards, resulting in a relatively large output voltage ripple that is considered unacceptable (Figure 9c).

This study adopts the Dickson structure as the voltage-doubling rectifier circuit due to constraints such as limited PCB board area, considering only four stages and below for the voltage doubler. The essential advantage of this circuit is that the output voltage escalates with an increase in the number of stages while maintaining a shallow DC voltage ripple. Given that the voltage regulator chip in the backend necessitates a turn-on voltage of 0.85 V, specific requirements are placed on the rectifier circuit’s output voltage.

As illustrated in Figure 10a, the initial fixed input voltage amplitude is set to 0.8 V. The impact of the number of Dickson structure stages and the load magnitude on both the output voltage and the efficiency of the rectifier circuit is assessed using simulation software. The rectifier circuit efficiency is quantified as the ratio of the DC output power of the voltage-doubling rectifier circuit at the load to the total power input at the circuit’s input, serving as a performance metric for the rectifier circuit. Subsequently, considerations are made regarding the output voltage, efficiency, and circuit area, leading to the selection of a three-stage Dickson voltage doubler with a load resistance of 3 kΩ.

For the three-stage Dickson voltage doubler, an analysis is conducted on the impact of the input voltage magnitude on both the output voltage and efficiency within the range of input voltages of interest. The outcomes of this investigation are delineated in Figure 10b.

First-order calculations can model any nonlinear voltage-doubling rectifier circuit with a load as an effective average load impedance (*R_in_*) using the following equations [40,41]:(4)$$R_{in}=\frac{V_{in}^{2}}{2\times P_{load}}\times \eta_{ac-dc}$$

*V_in_* is the peak value of the input AC voltage, *η_ac-dc_* is the efficiency of the rectifier circuit, and *P_load_* is the power consumed on the load. In this study, *V_in_* ranges from 0.4 to 1 V, *η_ac-dc_* ranges from 25.9 to 47.1%, *P_load_* ranges from 0.114 to 1.2 mW, and *R_in_* is calculated to be around 200 Ω. In the frequency range of 0.96–1 MHz, the voltage-doubling rectifier circuit and the piezoelectric receiver transducer exhibit similar resistances, allowing for efficient power transfer without needing a dedicated impedance-matching network. The regulator chip selected for this study is ADI’s LTC3400, a step-up DC/DC converter with a minimum turn-on voltage of 0.85 V, an output voltage range of 2.5–5 V, and a conversion efficiency of 92%. Supercapacitors are the preferred energy storage component for UWPT due to their nearly unlimited charging cycles and enhanced safety features compared to rechargeable batteries.

### 5.2. Implementation

The final designed circuit is implemented on a 15 mm diameter circular PCB with double-layer printing to conserve PCB area. The PMUT array is fixed on a PCB of the same size and electrically connected by bonding machine leads. The surface layer of the module is sealed with the acoustic matching gel adhesive with an acoustic impedance coefficient of 1.5 MRayl to waterproof the module and reduce the attenuation and losses caused by the propagation of acoustic signals between different mediums. The purpose of the sealing is to prevent waterproofing and to minimize the attenuation and loss of acoustic signals propagating between various media. The acoustic propagation coefficient at the interface of the two media can be expressed as follows [42]:(5)$$T=\frac{4Z_{1}Z_{2}}{{\left(Z_{1}+Z_{2}\right)}^{2}}$$
where Z_1_ and Z_2_ represent the acoustic impedance (expressed in MRayl) of the two media; the transmission coefficient between the acoustic matching gel adhesive and deionized water utilized in this study is 0.999. The back layer of the module incorporates transparent organic silicone to waterproof the circuit section. Notably, both sealants undergo a vacuum pumping process to remove residual air within the adhesive. They are subjected to heat to expedite curing, preventing soundwaves propagating within the adhesive from reflecting and attenuating upon encountering cavities.

Furthermore, a copper ring enclosure is employed to diminish the 50 Hz standard frequency interference within the experimental environment. The final wireless power-harvesting device (WPHD), illustrated in Figure 11a, showcases these design elements. In Figure 11c, the receiver module produces a DC voltage of 3.3 V in deionized water and is capable of illuminating an LED lamp. This demonstrates the module’s potential as a wireless power source for implantable medical systems.

## 6. Discussion

As shown in Figure 6c, the PZT transmitter demonstrates almost purely resistive characteristics when the signal generator produces an AC voltage at 980 kHz with a peak-to-peak value of 40 V. At this point, the output power density of the transmitter can be approximated using the following equation: (6)$$P=\frac{{\left(U_{p-p}/2\sqrt{2}\right)}^{2}}{R}\cdot \frac{1}{S}$$
where *R* is the equivalent impedance of the transmitter at the operating frequency, and *S* is the surface area of the transmitter. The incident power density is calculated as 705 mW/cm^2^, which is below the FDA limit of 720 mW/cm^2^. Given the water’s extremely low attenuation coefficient (0.002 dB/cm MHz), the power attenuation in the water can be neglected when the distance between the receiver and the transmitter is 2 cm. Therefore, the input power density of the receiver is approximately equal to the incident power density. Since the effective area of the receiver is 3.2 mm^2^, the input power is 22.56 mW.

Figure 7c,d show that the output power is maximized when the load resistance is 430 Ω. At this point, the peak-to-peak voltage across the load is measured as 1.345 Vpp, the power is calculated to be 525 μW, and the PTE is estimated to be 2.33%. Table 3 summarizes recent work on WPT devices, with traditional PZT materials being the most widely used in earlier years due to their excellent piezoelectric properties. Compared to the AlN materials commonly used in recent years, the AlScN used in this study enhances the material’s piezoelectric coefficient and optimizes the device’s receiving performance, which benefits the PTE.

In future experiments, conducting performance tests on the receiving transducer in an environment closer to human tissue, such as pork tissue, will yield results that more accurately reflect the device’s real-world performance. Further research will concentrate on more miniature and multi-directional piezoelectric transducers at the receiving end. The AlScN material’s CMOS compatibility allows for the integration of back-end circuitry and a PMUT array on a single chip, enabling extreme miniaturization of the device. Additionally, it is crucial to seal the entire device with a thin layer of parylene before implantation into the human body. Parylene provides exceptional stability and biocompatibility. Applying a thin layer of Parylene as a protective coating ensures optimal protection for both the body and the sensor without compromising the sensor’s functionality.

## 7. Conclusions

In summary, this study introduces a potential wireless energy charging approach within the human body, leveraging ultrasound-driven wireless energy transfer technology to convert acoustic energy into electrical energy using a centimeter-scale lead-free ultrasonic energy-harvesting device. The device utilizes AlScN-based PMUTs as the receiving transducers, and a series of characterization and testing experiments were conducted on them. The results showed that a maximum output power density of 16.44 mW/cm^2^ and a PTE of 2.33% can be generated at an input voltage of 40 Vpp. The rectified energy produced by the device was stored in a capacitor and converted into a stable DC output using a voltage regulator chip, successfully powering commercial LED devices. The ultrasound-driven wireless energy transfer system can potentially expand the range of applications for wireless energy charging.

## Figures and Tables

**Figure 1 micromachines-15-00624-f001:**
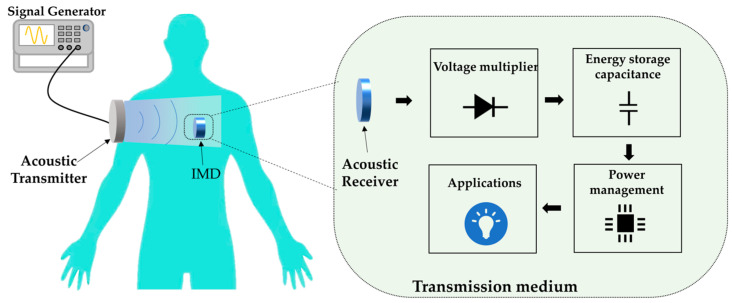
Diagram of the UWPT system framework.

**Figure 2 micromachines-15-00624-f002:**
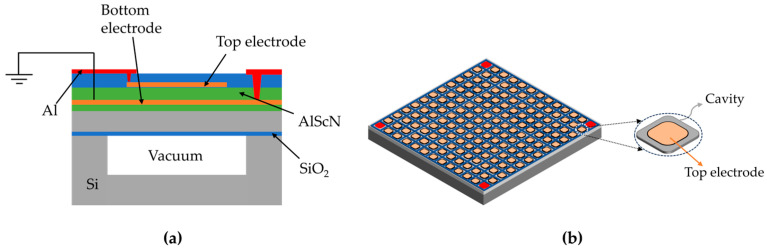
Array of PMUTs based on AlScN films. (**a**) Cross-sectional view of a PMUT. A piezoelectric film with AlScN on the silicon device layer is located on the vacuum cavity of the CSOI wafer substrate. (**b**) Perspective view of the 13 × 13 PMUT array and a magnified view of a single square sensing cell.

**Figure 3 micromachines-15-00624-f003:**
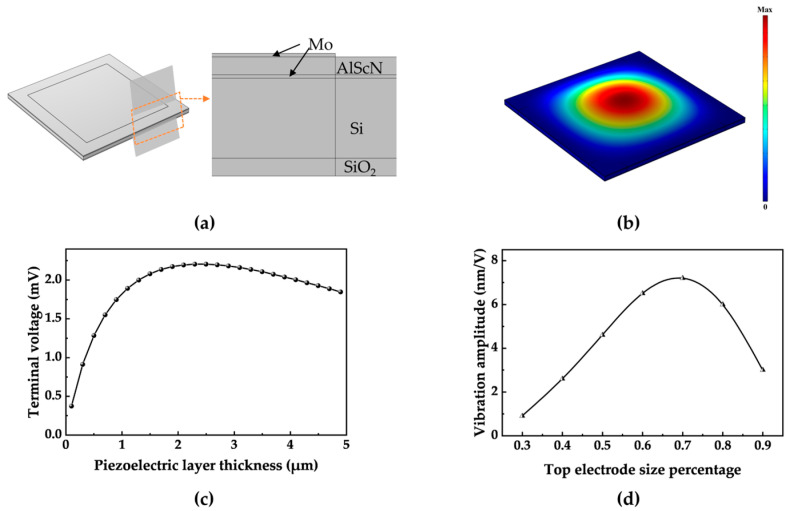
FEA simulation of the AlScN PMUT. (**a**) FEA simulation model of a PMUT and cross-section view. (**b**) First vibration mode shape of the PMUT. (**c**) Effects of different piezoelectric layer thicknesses on the output performance. (**d**) Piezoelectric film amplitude variations for different electrode size occupancies.

**Figure 4 micromachines-15-00624-f004:**
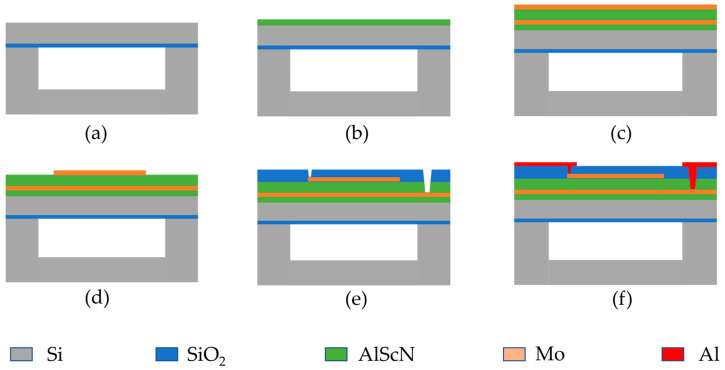
Flowchart of the PMUT manufacturing process. (**a**–**f**) Detailed fabrication process steps for ScAlN-based PMUT.

**Figure 5 micromachines-15-00624-f005:**
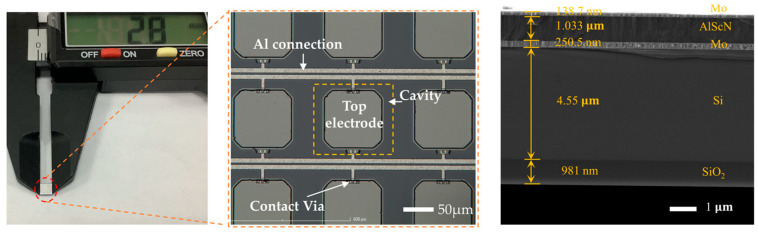
Close-up portraits of the PMUT array and an SEM cross-sectional image of the PMUT film layer.

**Figure 6 micromachines-15-00624-f006:**
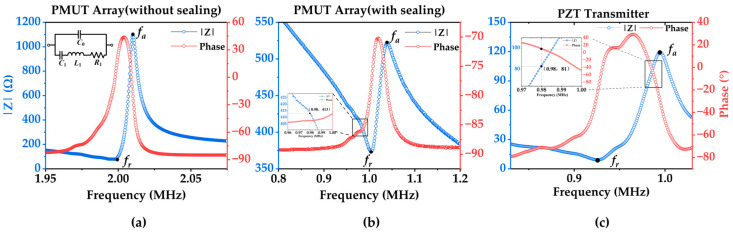
Results of impedance measurement experiments. (**a**) Impedance curve of the PMUT array without polyurethane adhesive sealing. The equivalent circuit diagram of the transducer is shown on the upper left. (**b**) The impedance curve of the PMUT array was sealed with polyurethane adhesive. (**c**) Impedance curve of the PZT transmitting probe.

**Figure 7 micromachines-15-00624-f007:**
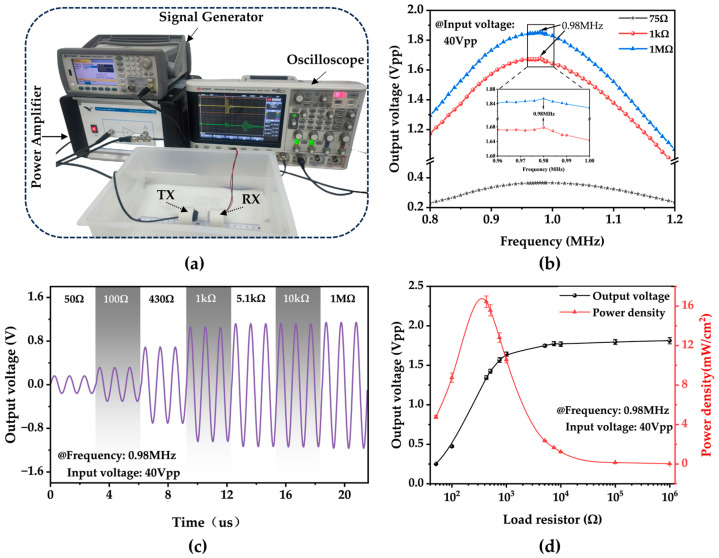
Test experiment of the PMUT array in deionized water. (**a**) Experimental setup. (**b**) Output voltage versus frequency curves of the transducer under different loads. The center is a local zoomed-in graph. (**c**) The output voltage amplitude of the transducer under various loads. (**d**) Variations in the output voltage and power density of the device with load, respectively, at 430 Ω for the optimal output power.

**Figure 8 micromachines-15-00624-f008:**
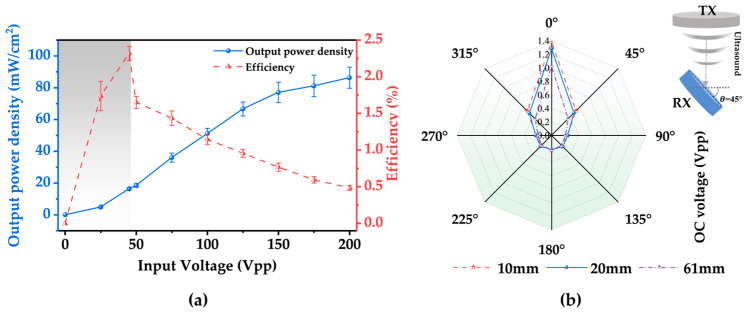
(**a**) Curves of the device output power density and transfer efficiency with the input voltage. (**b**) Variations in the device’s output voltage with the rotation angle showing the device’s ability to collect ultrasonic waves from different angles.

**Figure 9 micromachines-15-00624-f009:**
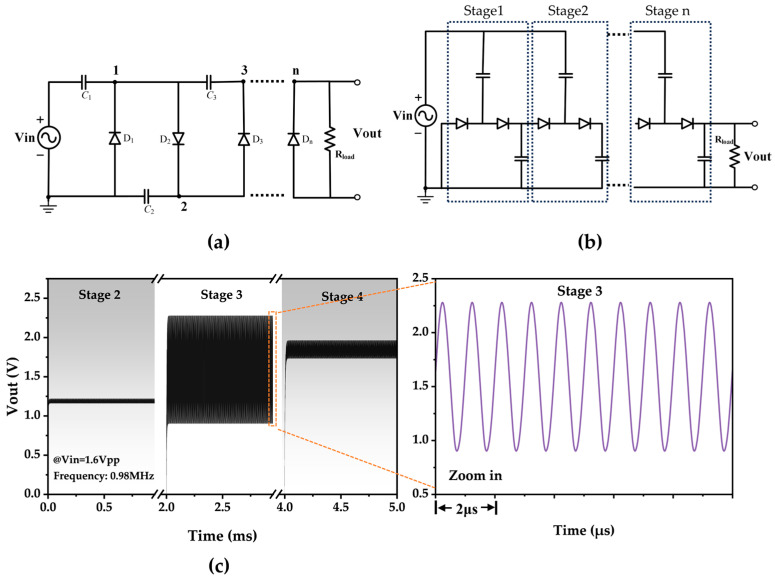
SPICE simulation results. (**a**) The basic circuit of the Villard voltage multiplier. (**b**) Basic n-stage circuit of the Dickson voltage multiplier. (**c**) Output voltages of 2–4 stages of the Villard voltage multiplier and local amplification of output voltages of the three-stage structure.

**Figure 10 micromachines-15-00624-f010:**
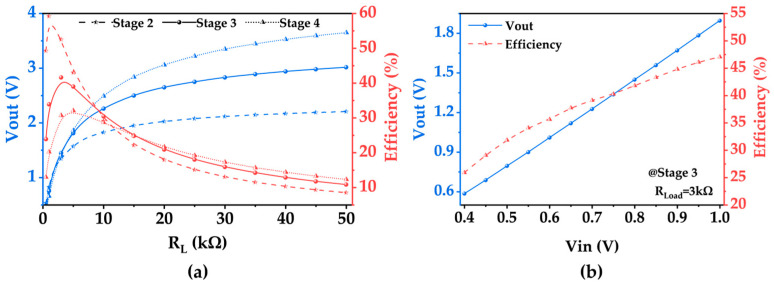
SPICE simulation results. (**a**) Output voltage and efficiency variation curves with the load for Dickson voltage multipliers of different configurations. (**b**) Output voltage and efficiency variation curves with the input voltage for a three-stage Dickson voltage multiplier.

**Figure 11 micromachines-15-00624-f011:**
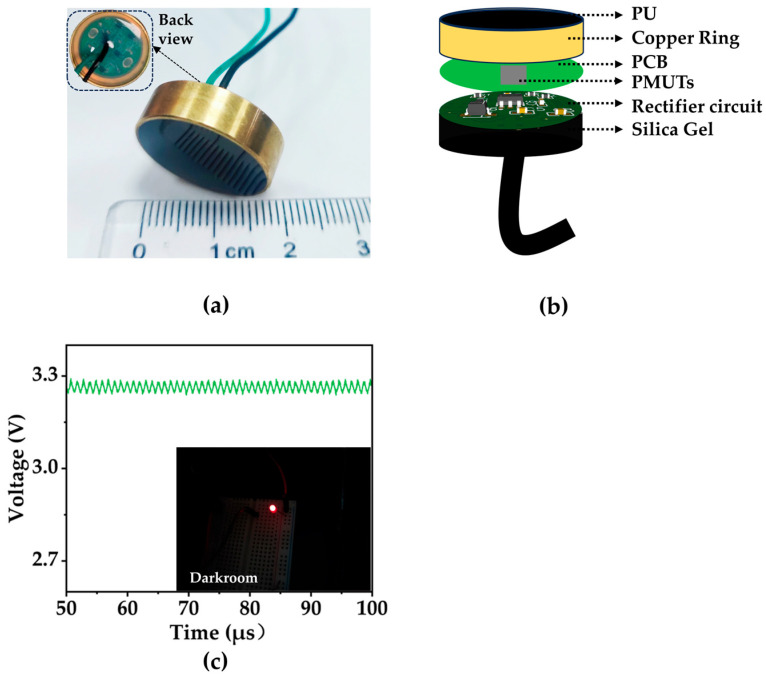
(**a**) Encapsulated WPHD. (**b**) Structural diagram of the WPHD. (**c**) The oscilloscope shows that the output voltage of the WPHD is ~3.3 V, and the LEDs are lit.

**Table 1 micromachines-15-00624-t001:** Comparison of material properties [24].

Property	AlN	Al_90.4%_Sc_9.6%_N
Piezoelectric coefficient, *e*_31,*f*_ (C/m^2^)	−1.06	−1.81
Relative permittivity, *ε*_33_ (F/m)	9.5	10.5
*e*_31,*f*_/*ε*_33_	−0.112	−0.172

**Table 2 micromachines-15-00624-t002:** Geometric size parameters of the PMUT.

PMUT Layer	Material	Side (μm)	Thickness (μm)
Top electrode	Mo	140	0.2
Piezoelectric layer	AlScN	-	1
Bottom electrode	Mo	-	0.2
Substrate	Si	-	4.5
Seed layer	SiO_2_	-	1
Cavity	-	180	15

**Table 3 micromachines-15-00624-t003:** Comparison of ultrasonic wireless power transmission schemes.

Year/Ref.	Receiver Material	Medium	Operating Frequency (kHz)	Transfer Distance (mm)	Input Power Density (mW/cm^2^)	Power to Load (μW)	Effective Area of Receiver (mm^2^)	PTE (%)
2014/[15]	PZT	Tissue	40.43	22	-	49	10.5	0.098
2019/[30]	PZT	DI water	88	20	322	-	4	0.33
2019/[43]	AlN	oil	2000	40	-	1	16	0.009
2019/[44]	AlN	Silicone oil	500	127	-	-	0.16	-
2022/[26]	AlN	PDMS	67	8.6	1	-	2.25	0.01
2022/[9]	AlN	DI water	3000	25	700	42	2.55	0.236
This work	AlScN	DI water	980	20	705	525	3.2	2.33

## Data Availability

Data are contained within the article.

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
