# Peer review of "An AlScN Piezoelectric Micromechanical Ultrasonic Transducer-Based Power-Harvesting Device for Wireless Power Transmission"

_micromachines, 2024, doi:10.3390/mi15050624_

Round 1

Reviewer 1 Report (New Reviewer)

Comments and Suggestions for Authors

This paper reports:

·       This work focuses on developing a miniaturized UWPT receiver device based on AlScN piezoelectric micro-electromechanical transducers (PMUT).

·       The proposed receiver features a PMUT array of 2.8 × 2.8 mm², comprised of 13 ×13 square elements. An acoustic matching gel is applied to address acoustic impedance mismatch when operating in liquid environments. Experimental evaluations in deionized water demonstrated that the power transfer efficiency (PTE) is up to 2.33%.

Comments:

This work demonstrates An AlScN piezoelectric micromechanical ultrasonic transducer based power harvesting device for wireless power transmission. This manuscript is not well organized and not well written. However, certain aspects must be discussed in depth before the work is published. Some of the serious concerns are given below.

·       In this work, authors used the AlScN as a piezoelectric material in PMUT array. Authors should provide the PFM response or piezoelectric coefficient of AlScN to further the confirmed the piezoelectric behavior of AlScN material.

·       Stability is an important factor to elucidate the performance of the energy harvesting. Authors should have mentioned the stability of the proposed TENG device.

·       The working mechanism of the AlScN piezoelectric micromechanical ultrasonic transducer is not properly explained. Authors should have explained in detail the working mechanism of the proposed device.

·       In this work, authors should calculate the signal to noise ratio of the AlScN piezoelectric micromechanical ultrasonic transducer based power harvesting device for wireless power transmission. Form the SNR, authors should explain that how much transmitted signal be modulated.

·       The reported PTE of 2.33% is comparitively low. Can authors explain that how the PTE can be improved for real-time applications or which techniques are required to improve the PTE?

·       Authors mentioned the thickness of AlScN piezoelectric material is 1 µm. Authors should explain the effect of the thickness and of AlScN piezoelectric material also effect of the device size on the output performance in the manuscript.

Comments on the Quality of English Language

none

Author Response

We thank the reviewer for the valuable comments and invite you to review more specific responses in the document below.

Reviewer 2 Report (New Reviewer)

Comments and Suggestions for Authors

Thanks for the well-written manuscript, I have the following questions for this paper:

1.       The author may give a thorough comparison of AlN and PZT's piezoelectric characteristics and overall performance efficacy in real-world PMUT applications. In particular, how do AlN-based PMUTs' sensitivity and signal-to-noise ratios in industrial or clinical settings compare to those of PZT-based PMUTs?

2.       The author may need to elaborate on possible strategies or technological developments that might mitigate the significant energy loss that high-frequency ultrasound transducers experience during propagation, which affects the efficiency of energy transmission. The energy loss must be avoided in order for devices like pacemakers to maintain the necessary depth of penetration. Which particular materials or waveform modulation methods have demonstrated potential for improving high-frequency transmission efficiency?

3.       The author may investigate the alternate media that would more precisely mimic the acoustic characteristics of soft tissue, considering the limits of deionized water due to its low attenuation coefficient in replicating human soft tissue environments. Furthermore, how do these substitutes measure up against deionized water and real human soft tissue in terms of their acoustic impedance and attenuation qualities for wireless acoustic energy transfer applications?

4.       Regarding Figure 7(a), where thermal effects and cavitation in liquids cause the output power density to plateau beyond an input voltage of 150 Vpp, what specific effects do these phenomena have on the efficacy and safety of ultrasound applications in medical settings, especially when interacting with soft tissues? Are there any set rules or limits for input voltage and power density as well, to maximize efficiency and safety (MI, TIS limits) in these applications?

5.       What are the expected challenges in reducing the size of these transducers about the creation of more tiny and multi-directional piezoelectric transducers at the receiving end? Moreover, how may the bidirectional properties improve these devices' usefulness and performance in real-world scenarios?

Author Response

We thank the reviewer for the valuable comments and invite you to review more specific responses in the document below.

Round 2

Reviewer 1 Report (New Reviewer)

Comments and Suggestions for Authors

This paper cannot be published in Micromachines as the authors did not take the reviewer's comments seriously. (Rejected)

Author Response

I deeply apologize for the previous response and I have made changes to the response. Please review the attached document.

This manuscript is a resubmission of an earlier submission. The following is a list of the peer review reports and author responses from that submission.

Round 1

Reviewer 1 Report

Comments and Suggestions for Authors

The author has clearly defined a theme. The author has done a lot of work on PMUT, and the final results look promising, but the overall article lacks some innovative points. Using PMUT based on AlScN material for energy harvesting systems may be an innovation, but I would prefer the author to show more creativity in other aspects. 

More detailed comments below:

1) Abstract, line 19: The author's intended meaning should be that the array size is 2.8 mm by 2.8 mm. There is ambiguity in the text, please rephrase.

2) Page 2, line 75: The word "coefficient" does not need a hyphen.

3) The introductory section does not reveal the novelty of the article and suggests some revisions.

4) Page 5, line 131: "acoustoelectric" is inaccurate and should be replaced.

5) The entire article makes no mention of the amount of Sc in AlScN.

6) Page 4, lines 160-167: The paragraph is vague in its explanation of the choice of operating frequency and does not explain well the reasons for the choice of 1 MHz.

7) No information was found on the dimensions of the PMUT structure.

8) Abstract line 21 and Pg 7, line 267: two references to output power compliant with FDA requirements, what does the FDA restriction on incident power density have to do with the output power of the implant?

9) Fig 7(b) conveys a confusing meaning, the upper right corner looks like an emission transducer, and no reasonable explanation is found in the article, suggesting additional clarification.

10) The article mentions "tissue loss" several times, but in the absence of relevant experiments in tissues, it is suggested to replace it with "medium loss".

Author Response

文档。review1“:对审稿人意见的逐点回复。

Reviewer 2 Report

Comments and Suggestions for Authors

The authors have identified a meaning topic and problem. The issue is that I cannot find the novelty of this work. There is the demonstration of lighting of LED that is nice to see. But in terms of knowledge, I am not sure what is new. There are also lots of flaws and misconceptions throughout the paper. The authors have worked aplenty on PMUT and are just porting over their pre-conceptions on PMUTs over to this domain of power harvesting. You cannot do that because once you do an analysis into the problem, you will find that the considerations are quite different. I would love to see more PMUT papers on power harvesting, but not in this form.  

More detailed comments below:

1) Abstract, line 19: this part is written in such a way that misleadingly says the elements are 2.8 mm by 2.8, when this size actually refers to the array. Please rephrase.

2) Abstract, line 21: The authors use the word "impressive" to describe their result. But there is nothing in the entire abstract provided to justify and contextualize this adjective. Such superlatives are meaningless in scientific writing. Instead, the authors should contextualize their findings for the scientific reader. And having read through the whole paper and examining Table I, I cannot see what is "impressive". Please enlighten me.

3) Pg 2, lines 71-74: The critique on PZT is naively simplistic. Please revise. 

4) Pg 2, line 76-82: The advocation of AlN is a simplistic textbook copy of the analysis from PMUTs. The problem of power harvesting and acoustic reception are quite different. The authors should do a proper analysis before coming up with such a claim. See later comments related to Table 1 for further substantiation.

5) Pg 2, line 75: The word "coefficient" does not have a hyphen.

6) The last paragraph of the introduction gives just a summary description of the device and what tests were done. What is the novelty of this paper?

7) Pg 3, lines 103-110: The author are once again thinking about this power generation problem in the simplistic way of a PMUT. It is not simply done to just resonance being the best. They have cited Arbabian's papers. Please read the papers and understand how they've done the analysis of impedance with respect to load matching and efficiency.

8) Fig 1: Where are these circuit elements (matching network, energy storage cap, etc) in this paper? This figure looks similar to other papers on this topic. If it is to frame the topic, given the existing coverage of the topic, there is really no need for this figure at all.

9) Pg 5, line 131: Where is the "acoustoelectric" effect in this entire paper? I do not see any acoustoelectric effect results in this paper. Do the authors actually understand this term? Have confused this term with piezoelectricity?

10) Pg 3, lines 139-140: This statement is flawed in numerous aspect. First, it's once again just copying over from a PMUT mindset. Second, how do the authors define receive sensitivity? Without a definition of receive sensitivity, it is not possible to validate how a higher fill factor affects the above figure of merit.

11) The authors mention AlScN in the fabrication section but there is no mention of the Sc content.

12) Pg 4, lines 160-167: This entire paragraph that I think the authors are trying to use to justify their choice of frequency is so convoluted to read. I am simply unable to follow. Basically the authors are trying to say there are multiple factor to consider on designing the frequency. That is obvious. But what then?

13) Pg 5, line 5: the authors mention thin receivers (<1mm) trade off against power efficiency. I am completely lost at the argument. How do the authors arrive at this conclusion. Is this argument even relevant to PMUTs? 1 mm is way off the scale of PMUT membranes and neither do I see how this could be related to the substrate thickness. It sounds like the authors have blindly copied over material related to bulk piezoelectrics.

14) There is zero information about the PMUT's lateral features: cavity size, top electrode size.

15) The authors mention tissue absorption multiple times as a consideration. What is the point when there are no through-tissue measurements? Please remove this term from the paper.

16) In relation to novelty, there is zero theoretical analysis (only vague misconceptions) on power generation to illustrate the merits of whatever the authors are proposing (which is a huge question mark).

17) Pg 5, line 188: what "through-holes" are the authors referring to? I also cannot see the point of this sentence about the various features apart from starting the obvious that the authors have a physical device. What is the point of saying the wire bonds are in tact?

18) Pg 5, line 199: why PMUT "arrays" (plural). I thought it was only one array (singular) comprising 13 by 13 elements. Just how many arrays are the authors using in their experiments? I am confused.

19) Equation (2): This is a classical mis-use of the equation based on resonance and anti-resonance points. The resonance of the PMUT, because it is just AlScaN, is too pathetically weak (in Fig 5b) for this equation to be admissible. The authors need to extract Cm and C0 and compute the coupling factor from the fitted parameters. Furthermore, the fact that Keff^2 increases with sealing should ring all the alarm bells that the authors doing something awfully wrong.

20) Fig 5a and 5c: Given the large charge in the impedance magnitude, could the authors plot the magnitudes on a log scale? I would like to see the spread in frequencies around fr. The impedances of the PMUT array looks unusually low for AlScN. Could the authors provide some validation against theoretical calculations?

21) Pg 7, line 267: The authors report the output power and the corresponding power density, then claim compliance with FDA regulations. But FDA regulations are on the limits of the incident power (in the case of medical ultrasound) and certainly not the output of an implant. I do not understand what the authors are trying to establish.

22) Fig 7a: Instead of plotting the input voltage to the transducer for the x-axis, it would be more relevant to plot the incident power density (where there is a limit by the FDA). There should also be a plot of the intrinsic efficiency of the power receiver (output power density / input power density). It would be interesting to compare this against the PZT bulk piezoelectrics published.

23) What is the disc in Fig 7(b)? I understand the TX transducer is a disc and it makes not sense to be showing the TX transducer as the arrow of the ultrasound is pointing towards the disc -- so this looks like a receiver. The receiver is supposed to be a square chip containing a PMUT array. This image is confusing. The spider graph showing the directional response is also confusing. What do the number labels mean? The output voltage? Why it is that voltage is highest at the furthest end along 0 degrees and is so tapered?

24) The authors show the simulation results but then jump to the validation with the LED. Where is the validation of the circuit (experiments vs simulation)?

25) Pg 12, line 409: The authors mention AlScaN for improving receiving sensitivity. As mentioned before, the authors are thinking about power harvesting as just like PMUTs, which is simplistic and unhelpful. Second, what is the relevance here? At the end, it is the power efficiency of that conversion. And the only FoM to compare here is the PTE. And the problem is the PTE of the AlN PMUT in reference [9] is 7 times of what the authors are demonstrating. 

26) Pg 12, line 414: again the authors annoyingly mention tissues when there are no tissues. Perhaps use "medium" instead here.

27) Table 1 is rather misleading in the way it lays out the materials. I think the architecture from one reference to another is not the same (i.e. not all are PMUTs). The architecture makes a difference. The authors should make a note someone on the architecture. Another issue is quoting input power. It should be input power density for normalization. The last two columns are pretty much redundant and should be removed.

28) Another insight from Table 1 is that it negates the claims on the benefits of AlScN for higher receive sensitivity using the typical PMUT argument. The PTEs are all way higher than what AlN or AlScN has produced. This should be pretty telling that the authors are looking at the wrong metric.

29) Pg 12, line 433: I cannot understand the authors' fascination with input voltages. After all the TX transducer is not the focus of this work. Instead, matters is the output power in relation to the input power density.

30) The abstract's mentioning of output power density and compliance to FDA limits echoes the confusion in the main text. The FDA limit is on the incident power. Having read through the whole paper, and then the abstract and conclusion again, I still cannot tell what the authors are trying claim. 

Author Response

Docs.“review2”: Point-by-point responses to reviewers' comments.

Reviewer 3 Report

Comments and Suggestions for Authors

This manuscript investigates a miniaturized UWPT receiver device based on AlScN piezoelectric micro-electromechanical transducer. Overall, the manuscript is impressive and well organized, which can be published after addressing the following issues.

Q1 What is the specific proportion of Sc doping and needs to be specified specifically.

For the AlScN thin film mentioned in the manuscript, it is recommended to supplement its XRD, SEM image, and transverse piezoelectric coefficient.

Q2 As the receiving end, what is the consideration for the design of the chip's microelement shape and the square array arrangement? What is the effect of array size on energy density?

Q3 What is the filling rate of the PMUT array in this manuscript, and how much has the receiving sensitivity been specifically improved?

Q4 In the sentence "the rotation angle of the RX exerts a considerable influence on the output voltage, while the distance also affects the output voltage to some extent. " What is the specific impact of rotation angle on output voltage, and is there a certain pattern?

Q5 Some content is repeated in different sections, it is recommended to consolidatethe information.

Q6 Why are the titles in Section 4.1 and 3.1 the structure of pmuts? Please carefully confirm the relevant contents of the article. EX:p.162-p.163

Q7 There are many types of polyurethanes, please specify the types of polyurethanes in the article, and explain in detail what is the basis for selection?

Q8 The language is generally good, but some sentences need improvement and some spelling errors need to be corrected.

Comments on the Quality of English Language

The language is generally good, but some sentences need improvement and some spelling errors need to be corrected.

Author Response

Docs.“review3”: Point-by-point responses to reviewers' comments.

Round 2

Reviewer 2 Report

Comments and Suggestions for Authors

The authors have no interest in considering my concerns. Apart from superficial corrections on typos, the most serious concerns have not been addressed. They clearly did not bother to read and consider these comments. Many of these concerns are flaws that the authors refuse to think about and correct. Some of the problems are fundamental conceptions that the authors need to think about instead of simply saying this will be done in future. The revisions and responses were clearly never read by the senior authors of this paper. More guidance and support need to be provided to the first author.

Author Response

Thank you for your professional advice, a more detailed response is in the document below.

Reviewer 3 Report

Comments and Suggestions for Authors

Consent to publication

Author Response

Thank you for your professional advice and your endorsement!